# Influence of Laser Energy Input and Shielding Gas Flow on Evaporation Fume during Laser Powder Bed Fusion of Zn Metal

**DOI:** 10.3390/ma14102677

**Published:** 2021-05-20

**Authors:** Yu Qin, Jinge Liu, Yanzhe Chen, Peng Wen, Yufeng Zheng, Yun Tian, Maximilian Voshage, Johannes Henrich Schleifenbaum

**Affiliations:** 1The State Key Laboratory of Tribology, Tsinghua University, Beijing 100084, China; qin-y17@mails.tsinghua.edu.cn (Y.Q.); liujg19@mails.tsinghua.edu.cn (J.L.); cyz76898@163.com (Y.C.); 2Department of Mechanical Engineering, Tsinghua University, Beijing 100084, China; 3Department of Materials Science and Engineering College of Engineering, Peking University, Beijing 100871, China; yfzheng@pku.edu.cn; 4Department of Orthopaedics, Peking University Third Hospital, Beijing 100191, China; tianyunbj@sina.com; 5Digital Additive Production (DAP), RWTH Aachen University, 52074 Aachen, Germany; maximilian.voshage@dap.rwth-aachen.de (M.V.); johannes.henrich.schleifenbaum@ilt.fraunhofer.de (J.H.S.)

**Keywords:** laser powder bed fusion, Zn, evaporation, numerical analysis, shielding gas flow

## Abstract

Laser powder bed fusion (LPBF) of Zn-based metals exhibits prominent advantages to produce customized biodegradable implants. However, massive evaporation occurs during laser melting of Zn so that it becomes a critical issue to modulate laser energy input and gas shielding conditions to eliminate the negative effect of evaporation fume during the LPBF process. In this research, two numerical models were established to simulate the interaction between the scanning laser and Zn metal as well as the interaction between the shielding gas flow and the evaporation fume, respectively. The first model predicted the evaporation rate under different laser energy input by taking the effect of evaporation on the conservation of energy, momentum, and mass into consideration. With the evaporation rate as the input, the second model predicted the elimination effect of evaporation fume under different conditions of shielding gas flow by taking the effect of the gas circulation system including geometrical design and flow rate. In the case involving an adequate laser energy input and an optimized shielding gas flow, the evaporation fume was efficiently removed from the processing chamber during the LPBF process. Furthermore, the influence of evaporation on surface quality densification was discussed by comparing LPBF of pure Zn and a Titanium alloy. The established numerical analysis not only helps to find the adequate laser energy input and the optimized shielding gas flow for the LPBF of Zn based metal, but is also beneficial to understand the influence of evaporation on the LPBF process.

## 1. Introduction

Laser powder bed fusion (LPBF), also referred to as selective laser melting (SLM), is an appropriate additive manufacturing method for fabricating metal bone implants of customized geometry, since it allows the precise melting of discrete fine metal powders layer upon layer under computer programming with high efficiency and high quality [1,2]. Bones are capable to repair themselves, and on many occasions, bone implants are expected to degrade gradually and dissolve completely with the reconstruction of new bones [3,4]. Zn plays a vital role in bone metabolism and has a moderate corrosion rate in the human body, which tends to match the bone healing rate, as compared to fast-degrading Mg or slowly degrading Fe [5,6]. The strength of pure Zinc is not enough for load-bearing implants. Yang et al. made a comprehensive study on the biodegradable application of Zn alloys, and both in-vitro and in-vivo results were quite promising regarding mechanical strength, biocompatibility, biodegradation, and osteogenesis [7]. Considering the customized geometry of bone implants, LPBF of Zn-based metals (pure Zn and Zn alloys) has also been attempted [8,9,10]. The Young’s modulus of pure Zn porous scaffolds produced by the LPBF decreased to as low as 0.7–1.0 GPa, which fell within the range of trabecular bones, and the strength did not decrease even after 4 weeks of immersion in revised simulated body fluid [9]. Therefore, the LPBF of biodegradable Zn-based metals is expected to solve the dual technical difficulties, including the fact that “conventional manufacturing processes are inadequate to fabricate personalized implants of complicated structure” and “conventional medical metals are biologically inert and exist in the human body permanently”. It may provide a novel clinical treatment for large bone defects [11].

Evaporation is a very significant phenomenon during the LPBF process considering the utilization of a focused laser beam with high energy intensity over 10^6^ W/cm^2^, fine powder of 20~60 μm, and a long running time of hours in a closed chamber. With X-ray visualization of the molten pool, Cunningham et al. found out that strong evaporation actually occurs in most LPBF processes, which was proved by the formation of a keyhole resulted from the recoil force of evaporation [12]. The keyhole, also named vapor depression, is a cavity of metal vapor inside the metal caused by the movement of the liquid/vapor interface and driven by the recoil force of evaporation. Khairallah et al. pointed out that the keyhole behavior directly influenced the absorption of laser energy and the formation of defects such as porosity and spatter during the LPBF process [13,14]. Verhaeghe et al. found that the temperature evolution was considerably affected by the heat loss from evaporation, explained by the fact that the latent heat of evaporation is dozens of times that of fusion [15]. Wu et al. found that the calculated peak temperature of the molten pool was considerably reduced by taking the recoil force of evaporation into account, since the recoil force accelerated the fluid flow and enhanced the heat convection in the molten pool [16].

Klassen et al. established a model to predict the element loss due to evaporation during the electron beam melting process [17,18]. The condensation flux associated with the formation of a Knudsen layer can significantly reduce the vaporization mass flux, thus also the recoil pressure. With considering the Mach number of the vapor at the Knudsen layer, they used the coefficients 0.82 and 0.56 for the evaporation and recoil pressure for electron beam powder bed fusion. When the evaporation rate of the contained alloying elements is varied, the chemical compositions of the produced parts become very different from the starting powder, namely with more loss of elements that have relatively high evaporation rates [19,20,21]. The metal vapor cools down and forms fumes of very fine particles in the closed processing chamber. The evaporation fume blocks the irradiation of the laser beam and attenuates the absorbed energy on the powder bed, which results in deteriorated formation quality. Ferrara and Ladewiga et al. made use of the shielding gas flow to eliminate the evaporation fume, and numerically investigated the influence of gas circulation system on the shielding gas flow [22,23]. They pointed out that a forceful and uniform shielding gas flow was beneficial to eliminate the evaporation fume and to stabilize high formation quality. However, the effect of laser energy on the evaporation fume has not been considered so far. With increasing the laser energy input, higher temperature and more evaporation are expected. Few studies have been found on how to design the shielding gas flow based on the volume of the generated evaporation fume under different laser energy input.

Besides the laser energy input, the evaporation volume is also determined by the evaporation tendency of materials. The melting and boiling temperature of pure Zn are as low as 420 and 907 °C, respectively. Zn is much more likely to evaporate than most metals due to the relatively low boiling temperature and narrow temperature range between fusion and evaporation. Montani et al. made an early study on the LPBF of pure Zn [24,25,26]. Massive evaporation fumes occurred during the laser melting of pure Zn powder and the highest relative density was only 88% [24]. The relative density indicates the densification degree and can evaluate the percentage of formation defects like pores and insufficient fusion. Grasso et al. found that the formation of porosity was directly related to the evaporation fume by in-situ infrared imaging [25]. Stable melting during the LPBF of Zn powder could be achieved only when the negative effect of the evaporation fume was eliminated [26]. The evaporation fume was efficiently blown away and suctioned out from the processing chamber by a customized gas circulation system during the LPBF of pure Zn [27]. The relative density remained stable above 99.5% and the tensile properties of LPBF pure Zn parts approached those obtained by hot extrusion [28]. The evaporation becomes the most critical issue accordingly regarding the processing and quality control during the LPBF of Zn-based metals. So far, the question remains of how to accurately evaluate the influence of the laser energy input and the shielding gas flow on the evaporation fume for the LPBF of Zn-based metals, as well as how to efficiently select the optimal parameters to achieve stable formation quality.

In this paper, the evaporation rate under different laser energy input was calculated during the LPBF of pure Zn. The interaction of the shielding gas and the evaporation fume was simulated using the calculated evaporation rate as the input. The effect of Zn evaporation on the melting and densification behavior was characterized, and compared with the LPBF of TC4 titanium alloy (Ti6Al4V), which has been widely used as the material of non-degradable metal implants and shows a much lower evaporation tendency. It is important to understand the influence of laser energy input and shielding gas flow on the evaporation fume during the LPBF of Zn metal and efficiently achieve a stable formation quality to promote the application of biodegradable Zn bone implants produced by the LPBF.

## 2. Numerical Simulation Methods

### 2.1. Numerical Modeling of the Interaction between the Laser and Metal

#### 2.1.1. Conservation Equations and Assumptions

A numerical model was established to simulate the interaction between the laser and metal by using the commercial software ANSYS Fluent 16.0. The calculation was performed in a domain with dimensions of 1.5 × 0.6 × 0.5 mm^3^. The domain was meshed into hexahedral elements sized 0.01 × 0.01 × 0.01 mm^3^. The finite volume method was utilized to solve the three conservation equations, i.e., the energy, momentum, and mass [29].


(1)$$\mathrm{Energy}\;\mathrm{equation}:\;\rho \left(\frac{\partial T}{\partial t}+\vec{V}\cdot \nabla T\right)=\nabla \cdot \left(\lambda \nabla T\right)+S_{E}$$



(2)$$\mathrm{Momentum}\;\mathrm{equation}:\;\rho \left(\frac{\partial \vec{V}}{\partial t}+\vec{V}\cdot \nabla \vec{V}\right)=\mu \nabla^{2}\vec{V}-\nabla p+M_{s}\cdot \vec{V}+S_{F}$$


(3)$$\mathrm{Continuity}\;\mathrm{equation}:\;\frac{\partial \rho }{\partial t}+\nabla \cdot \left(\rho \vec{V}\right)=S_{M}$$
where *λ*, *ρ*, *μ*, and *p* indicate the heat conductivity, density, viscosity, and pressure; t and T denote the time and temperature, respectively. $\vec{V}$ represents the velocity of the liquid metal, *S_E_* is the energy source, *S_F_* is the force source, and *S_M_* is the mass source.

In this calculation, gas/liquid/solid phases were coupled, and a moving heat source was considered. A solid metal plate, rather than powder bed, was used in this model based on the following reasons. Firstly, the mass loss of the Zn powder caused by the evaporation is difficult to evaluate experimentally. Secondly, the present powder scale model is regarded as inappropriate under conditions of intense vaporization [13]. Last but not the least, the laser energy input and the material properties are the driving force of the evaporation, while the presence of a layer of powders results in only a second-order effect [12]. The following simplification and assumptions were also applied to the numerical model according to reference [13].

(a) The powder bed was treated as a dense continuum with isotropic properties.

(b) The fluid model was treated as incompressible Newtonian flow. 

(c) The atmosphere in the LPBF chamber was pure argon with the temperature of 300 K and the pressure of 1.07 bar.

(d) The influence of the Knudsen layer on the gas parameters near the keyhole was omitted.

(e) The interaction between the metal vapor and the incoming laser beam was omitted.

#### 2.1.2. Treatment of Source Items and the Interface between Liquid and Gas 

If the temperature of the molten pool exceeded the boiling point *T_b_*, the evaporation occurred and led to heat loss, recoil force, and mass loss, all of which were considered in source items and were coupled in the iterative computation. For the energy transfer, *S_E_* was calculated considering the efficient energy input from the laser source (*S_q_*) and the latent heat from the phase transformation (*S_H_*). The heat flux from the laser was distributed in the form of the Gaussian distribution at the planes perpendicular to the z-axis, and was simulated by a three-dimensional rotating body heat source, as given in Equation (4) [30,31]. Once the temperature exceeded the boiling point, the additional energy input contributed to vaporize a fraction of the metal. Therefore, it was reasonable to assume that the peak temperature in the molten pool was just a bit higher than the boiling point for a stable melting during the LPBF. The mass fraction of the vaporized material was calculated from the temperature range beyond the boiling point, as expressed in Equation (5) [32]. With knowing the temperature and the evaporation mass, the latent heat *S_H_* including solid to liquid and liquid to gas was considered in the numerical model, as expressed in Equation (6).

For the momentum transfer, *S_F_* was calculated considering the recoil pressure *P_r_*, the surface tension *P_σ_*, and the buoyancy force *P_B_*, as expressed in Equations (7)–(10) [30,31]. For *S_M_* in the mass transfer, the change of mass *m*′ was a result of the evaporation between the liquid and gas. The gain in the mass of gas equaled the loss in the mass of the liquid in the form of evaporation. During the LPBF, the laser moves so fast that the computation domain needs to be large enough to achieve the constringency, which consumes enormous computing resources. To improve calculation efficiency, the laser beam was fixed in the coordinate system, and the work piece moved relatively to the laser beam as the scanning speed. The source terms related to the scanning were also incorporated into the model.
(4)$$q\left(x,y,z\right)=\frac{9\eta P}{\pi hr^{2}\left(1-e^{-3}\right)}exp\left[\frac{-9\left(x^{2}+y^{2}\right)}{r^{2}\ln\left(\frac{h}{z}\right)}\right]$$
(5)$$m^{\prime }=\frac{c_{p}\left(T-T_{b}\right)}{H_{b}}\cdot \frac{m_{e}}{dt}$$
(6)$$S_{H}=\rho \left(\frac{\partial }{\partial t}H_{m}+\nabla \cdot \left(\vec{V}H_{m}\right)\right)+m^{\prime }H_{b}$$
(7)$$P_{r}=0.54P_{0}exp\left(H_{b}\frac{T-T_{b}}{RT_{b}T}\right)$$
(8)$$P_{\sigma }=k\gamma$$
(9)$$\gamma =\left(1.557-1.5\cdot 10^{-4}\cdot \left(T-T_{m}\right)\right)$$
(10)$$P_{B}=\rho g\varphi \left(T-T_{m}\right)$$
(11)$$\frac{\partial F}{\partial t}+\vec{V}\cdot \nabla F=0$$

The free surface between liquid and gas in the computation domain was constructed by the Volume of Fraction (VOF) method as Equation (11) shows. F is the function indicating the phase status. The gradient of the volume fraction was obtained by comparing the calculated F values of adjacent elements, which distinguished the free surface between liquid and gas. All the source items were written by C coding and put into the mode by user-defined functions (UDF) [30,31]. The thermal physical properties employed in this work were presented as the Appendix A.

### 2.2. Numerical Modeling of the Interaction between the Shielding Gas and Evaporation Fume

Figure 1 and Figure 2 respectively show the geometrical design and numerical model of the LPBF processing chamber. They were in the same dimensional size of 150 × 275 × 100 mm^3^. A gas circulation system was utilized, in which shielding argon gas was fed into the chamber through a blow-off inlet and pumped out through a suction outlet. Ventilation screens were used to adjust the shielding gas flow. For the blow-off screen, small holes with a diameter of 1 mm were vertically staggered to rectify the gas flow. The suction screen was shaped like a horn mouth to force the flow and to collect the evaporation products. The effect of the geometrical design of ventilation screens on the shielding gas flow had been numerically simulated in previous studies [33] and was thus not discussed here. The performance of the shielding gas flow was determined by the velocity of gas flow at the inlet (*V_b_*) and the suction pressure at the outlet (*P_s_*). Figure 2b shows the numerical model after meshing. Two million hexahedron elements were used, and the element size exhibited a graded distribution. The minimum element size was set to 1 mm at the gas inlet, outlet, and evaporation zone, and the size was increased to 4 mm gradually in order to get balanced performance between calculation accuracy and efficiency. 

The interaction between the shielding gas and the evaporation fume was simulated in three steps. At first, the evaporation rate was calculated according to the interaction between the laser and the metal. The laser energy input and material properties were set as the input variables. Then, the flow behavior of the shielding gas inside the processing chamber was calculated. The parameters of gas flow *V_b_* and *P_s_* were set as the input variables. Finally, the evaporation fume was released from the bottom of the processing chamber as the mass flow according to the evaporation rate. ANSYS Fluent 16.0 was used for the simulation of the gas flow and the mass flow. The k-epsilon standard model was used to solve the Reynolds-averaged Navier–Stokes (RANS) equation.

## 3. Materials and Experiments

Pure Zn plates with a thickness of 25 mm were used to verify the calculation accuracy by comparing the penetration shape and the evaporation mass loss. A customized LPBF machine was used to perform laser melting in a closed chamber as Figure 1 shows. The optical system consists of a single mode continuous ytterbium fiber laser (YLR-400, IPG, Oxford, MA. USA) with the laser spot diameter of 75 μm and the maximum power of 400 W at wavelength of 1070 μm, a galvanometric scanner (hurry SCAN 20, SCANLAB, Puchheim, Germany) and a f-theta focusing lens (SILL S4LFT 3254/126. Sill Optics, Wendelstein, Germany). The laser was guided into the processing chamber through a transparent mirror and selectively melted the pure Zn plates on the platform. The gas atmosphere of the chamber was kept in pure Argon (>99.999%), and was continuously monitored by an oxygen meter (<100 ppm). More information can be found in our previous work regarding the LPBF process [27,28].

Laser melting was performed on three pieces of pure Zn plates in the LPBF processing chamber. The spot diameter of laser was 75 μm; the scanning speed was fixed as 500 mm/s; and the laser power was varied as 70, 80, and 90 W. Parallel single tracks with a length of 25 mm were melted on the pure Zn plates. The offset between two tracks was set as 200 µm to ensure that no overlap occurred between the tracks. It took 1.75s to melt 35 tracks in the total length of 875 mm. The evaporation rate equaled the mass loss divided by the scanning time. Figure 3 shows the Zn plates before and after the laser melting. The plates were weighed using a precision balance (±0.1 mg) to calculate the mass loss after the melting. The measurement was repeated 3 times under each condition to obtain the average value. Ultrasonic cleaning was performed to remove any impurities before melting and weighing. The cross-section of a single track on the pure Zn plate was cut, polished, and etched to observe the penetration shape. A solution of 2 mL nitric acid and 100 mL deionized water was used as the etchant. 

## 4. Results and Discussion

### 4.1. Penetration Shape

Figure 4 shows the comparison of the penetration shape in the Zn plates and the calculated temperature contours corresponding to the laser powers of 70, 80, and 90 W. Porosity was not observed at the cross-sections of the laser melted plate, indicating a satisfactory formation quality. The penetration shape agreed with the calculated isothermal line of the melting point. With the elevated laser power, the depth and width increased owing to the increased energy input, as shown in Figure 5. The penetration shape changed from a semi-ellipse to a pyramid cone. The depth-to-width ratio of the penetration was 0.34, 0.54, and 0.67 for a laser power of 70, 80, and 90 W, respectively. The penetration depth increased more rapidly than the width as the depth was more sensitive to the increase of laser power. 

Besides the laser power, the material properties also influenced the penetration shape. Figure 6 shows the comparison between the calculated temperature contours for TC4 and pure Zn under the same laser energy input. The isothermal line of the melting point indicates the penetration shape, the velocity vector indicates the flow behavior inside the molten pool, and the length of the vector indicates the magnitude of the velocity. The liquid metal in the molten pool flowed from the center to the surrounding and from the surface to the bottom for both the materials. Compared with the flow velocity for TC4, the flow velocity of pure Zn was considerably higher. A strong downward flow was observed for pure Zn, and consequently, the molten pool of pure Zn was wider and deeper than those of the TC4. 

### 4.2. Evaporation Rate

Pure Zn plates were melted continuously in the LPBF processing chamber. During the melting process, the evaporation fume was suctioned out by the gas circulation system, resulting in mass loss of the plates. The evaporation rate was determined as the mass loss divided by the scanning time. Table 1 presents the evaporation rate under different laser power, as obtained by measurement and calculation. Based on the experimental measurement, the corresponding evaporation rates were 0.23, 0.29, and 0.57 mg/s for laser powers of 70, 80, and 90 W, respectively. With enhancing laser power, the evaporation rate increased. A substantial increase occurred when the laser power went up to 90 W. The same tendency was noted in the calculated values. The largest deviation in the evaporation rates obtained using the measurement and calculation was approximately 10%. The calculated evaporation rates exhibited satisfactory accuracy.

According to the theoretical analysis based on the Langmuir model in Equation (12) [34], the evaporation flux of element i of a certain alloy *J_i_* (g·cm^−2^·s^−1^) is dependent on the temperature of the molten pool *T*, the saturated vapor pressure *P*_i_^0^, and the mole mass *M*_i_. As Figure 7a shows, the *P*_i_^0^ of the pure element increased with increasing the temperature according to equation 13 [35]. Pure Zn shows the highest saturated vapor pressure. As Figure 7b shows, *J*_i_ also increases with increasing the temperature. Thus, the evaporation loss of alloying elements rises with increasing the laser energy input, which causes an increased temperature in the molten pool. The evaporation loss tendency of different materials ranks as Zn > Mg > Fe > Ti. Besides the evaporation flux, the compositional change of a specific alloy is also related to the original concentration of an alloying element. For example, a lower concentration tends to lead to a higher relative loss regarding the similar evaporation rate [34]. Moreover, the Langmuir model reveals the maximum evaporation loss during melting in the vacuum. Actually, there is a Knudsen layer with several mean free paths at the surface of the molten pool during the LPBF, which is not in translational equilibrium [36]. The Knudsen inhibits the continuous evaporation, and the condensation of metal vapor may fall back to the molten pool, both of which reduce the evaporation loss in total [37,38].
(12)$$J_{i}=\frac{P_{i}^{0}}{\sqrt{2\pi M_{i}RT}}$$
(13)$$\log P_{i}^{0}=AT^{-1}+B\log T+C$$

### 4.3. Effect of the Shielding Gas on the Evaporation Fume 

Figure 8 shows the flow behavior of the shielding gas in the processing chamber, as obtained by the simulation with *V_b_* = 0.75 m/s and *P_s_* = −500 Pa. The argon gas was fed into the processing chamber through a blow-off screen at the right side and suctioned out at the left side by pumping. According to the calculation, the velocity of the gas flow adjacent to the powder bed was as low as 0.3 m/s. The low velocity near the powder bed is necessary for LPBF processing as a strong gas flow may denudate the powder. The velocity was enforced at a height ranging from 5 to 20 mm, and no significant turbulence was observed throughout the chamber. The maximum velocity was 0.88 m/s, and it appeared near the blow-off inlet. The minimum velocity was 0.74 m/s, and it occurred near the suction outlet at the two sides. The velocity contour indicated that a homogeneous gas flow field was achieved, and the experimental measurement validated the calculation accuracy with the biggest error less than 10% for the velocity [33]. 

A coupled model was established to predict the effect of the shielding gas on the evaporation fume during the LPBF of pure Zn by considering both of the evaporation rate, which is dependent on the laser energy input and material properties, and the shielding gas flow, which is dependent on the gas circulation system and gas flow parameters. An inlet of mass flow was added at the bottom of the processing chamber to simulate the evaporation fume. The inlet mass flow was set as the evaporation rate. Three different typical shielding conditions were simulated based on experimental results: *V_b_* = 0 and *P_s_* = 0 indicating no shielding gas flow; *V_b_* = 0.4 m/s and *P_s_* = −500 Pa representing the insufficient shielding gas flow; *V_b_* = 0.75 m/s and *P_s_* = −500 Pa expressing the adequate shielding gas flow. The calculation results are shown in Figure 9, and the corresponding videos are presented as Appendix A.

According to Figure 9a, the evaporation fume erupts and rises directly when no shielding gas flow is used. Shortly, the closed processing chamber is occupied by the saturated evaporation fume, which was observed by Montani and Grasso et al. during the LPBF of pure Zn [24,25]. When the insufficient shielding gas flow is used as shown in Figure 9b, the evaporation fume is blown away. A part of the evaporation fume is suctioned out, and the residual remains in the chamber. With increase in the scanning time, the residual evaporation fume enhances and accumulates inside the chamber. Figure 10a shows a captured image of the evaporation fume during the LPBF of Zn powder with the same setting of the shielding gas flow. The movement of the evaporation fume is similar to that observed in the simulation result. The residual evaporation fume cools down and gradually condenses into small particles that aggregate to different orders of scales. The particles scatter the irradiation of the laser beam, and attenuate the real energy input on the powder bed, which leads to an unstable molten pool and high porosity. Moreover, some of the overheated particles may reach the transmission mirror at the top of the processing chamber. In the extreme case, the mirror may get damaged owing to thermal distortion resulting from the attached smudges.

When the adequate shielding gas flow is used, as shown in Figure 9c, the evaporation fume is efficiently removed from the processing chamber. No residual evaporation product exists in the atmosphere inside the chamber with the passage of time. Figure 10b shows the captured image of the evaporation fume during the LPBF of Zn powder under the same shielding conditions. The simulated movement of the evaporation fume corresponds well with the experimental observation. Under suitable shielding conditions, a clean processing atmosphere is maintained during the LPBF process even though the strong evaporation fume erupts. Further enforcing the shielding gas flow is not recommended since it leads to a turbulent gas flow that may result in a disordered distribution of the flow velocity. Moreover, the appropriate shielding condition should be selected depending on the laser energy input and material properties. 

### 4.4. Discussion

Evaporation influences the penetration shape during the LPBF primarily in two aspects. On one hand, the particles inside the evaporation fume attenuates the laser energy input on the powder bed, and results in unstable penetration shape. Stable melting and high densification can be obtained only if the evaporation fume is removed efficiently from the processing chamber [26]. On the other hand, the recoil force of the evaporation leads to the occurrence of the keyhole, which considerably elevates the absorptivity of the laser energy and results in a deep penetration [13]. Figure 6 shows the comparison between the penetration shapes of TC4 and pure Zn under the same laser energy input without considering the attenuation effect. Zn owns much higher saturated vapor pressure and more evaporation compared to Ti as Figure 7 shows. The strong recoil force of the evaporation drives the liquid metal away from the laser-metal-interaction zone to the surrounding and results in a wider and deeper penetration for the LPBF of pure Zn. The downward flow is particularly strong, which considerably increases the penetration depth.

The evaporation rate rises up with the increase of the temperature of the molten pool. With increasing the laser energy input, the temperature of the molten pool rises up sharply. However, when the temperature exceeds the vaporization point, massive evaporation occurs and the temperature cannot continue to rise up. The increased laser energy will cause more evaporation, and the evaporation will carry excess heat away from the molten pool. Therefore, it is reasonable to assume that the temperature of the molten pool is a bit higher than the vaporization point, and many simulation results have proved this assumption [13,14,32]. In our model, the peak temperature of the molten pool was set to the vaporization point and the evaporation loss was considered according to equation 5. However, this method cannot reflect the compositional change of different elements for alloys. Klassen et al. simulated multi-component evaporation during electron beam powder bed fusion by using the Langmuir model, which was appropriate for the melting in vacuum [17,18]. Debroy et al. proposed to simulate the composition change resulted by evaporation loss with considering the effect of Knudsen layer during laser welding [37,38]. The total evaporation loss included two parts: One part driven by the excess pressure and the other part caused by diffusion. This model was appropriate to consider conduction mode laser welding only [39]. According to recent findings, the keyhole generally exists during most LPBF process [12]. So far, most LPBF models have paid more attention to the effect of recoil force due to the evaporation on the melt flow, rather than the compositional change since the evaporation loss during the LPBF of commonly used metals are considered not noticeable [13]. However, regarding the LPBF of biodegradable Zn and Mg alloys, the massive evaporation loss of Zn and Mg element caused substantial compositional change [11]. More work is expected in the future for the numerical simulation of compositional change during the LPBF process.

The metal vapor cools down and turns into evaporation fume that consists of tiny particles with a diameter at 10–150 nm, which plays a detrimental role in scattering the laser beam and contaminating the processing chamber [40]. Customized shielding gas flow system, including blow-off and suction-out gates, was used to eliminate the evaporation fume [22,23]. For the LPBF of Zn-based metals, the elimination effect contributed by the shielding gas flow determines the formation quality due to the occurrence of massive Zn evaporation [27,28]. As Figure 8 shows, powerful and uniform shielding gas flow was obtained. It was necessary to keep the height of the gas flow some distance away from the powder bed. The action region was kept 5 mm away from the powder bed and lasted for a range of 15 mm. If this distance was too low, the gas flow disordered the stacked powder, and disturbed the fluid flow inside the molten pool. If the distance was too high or the range was too small, the gas flow missed the evaporation fume and weakened the elimination effect. If the range was too big, it became difficult to maintain a uniform distribution of powerful gas flow. It was necessary to adjust the shielding gas flow according to the processing condition and the used materials. As Figure 9b shows, if the shielding gas flow was insufficient, the residual evaporation fume spread into the upper space of the processing chamber and gradually disturbed the melting process.

In order to further understand the effect of evaporation, the LPBF of pure Zn powder was carried out with efficiently eliminating the evaporation fume. Figure 11a,b show surface images of single tracks with laser power 80 W under different scanning speeds from left of right: 200, 400, 600, 800, and 1000 mm/s [28]. Plenty of molten liquid was ejected out from the molten pool due to the severe evaporation. They fell down under gravity and transformed into spherical balls due to surface tension. The ejection disturbed the adjacent powders, and also pushed them away from the molten pool. The solidified spatters and partially melted powders attached to the surface of single tracks. The molten pool showed violent movement due to the recoil force of evaporation. Under the rapid cooling rate, the turbulent molten metal solidified with a rugged surface and twisted shape. Figure 11c shows the top surface of pure Zn cubes built by the LPBF. Although the surface quality of single tracks was poor, the surface quality of pure Zn cubes was improved and comparable to those obtained by the LPBF of common metals. A lot of overlap melting occurred accordingly when the powder bed was melted track by track and layer by layer during the LPBF process. The numerous melting and wetting produced a smooth surface and enhanced the surface quality.

The processing window of high density over 99.5% was obtained in our previous work [27], represented by the gray area in Figure 12. The optimal linear energy input (laser power *P* divided by scanning speed *V*) was in the range of 127–289 J/m. A much lower energy input could not melt the powder completely and resulted in the lack of fusion. In contrast, a much higher energy input aggravated the evaporation and led to a high porosity owing to the gas entrapment. Nevertheless, when *P* was more than 120 W, too much evaporation occurred, and a high densification could not be obtained. Therefore, a relatively low *V* was necessary for the LPBF of pure Zn, not only to provide enough energy to melt the powder, but also to suppress the intense evaporation. Figure 12 also shows the optimal laser energy input for the LPBF of TC4 powder for the comparison [41,42,43,44,45,46]. The orange region indicates that the optimal energy input was 182–960 J/m, which was considerably larger than that of pure Zn. The combination of high *P* and *V* or low *P* and low *V* both led to the achievement of a high densification in the case of TC4, but the combination of high *P* and high *V* did not work in the case of pure Zn. The different densification behavior between Zn and TC4 is hugely attributed to the different evaporation tendency. The evaporation of Zn is so prominent that it is difficult to maintain sufficient melting and minimum evaporation at the same time when high *P* and high *V* are used. 

## 5. Conclusions

Addressing the evaporation and its resulting problems during the LPBF of Zn and its alloys is a critical issue. To this end, the effect of evaporation on the LPBF process was numerically simulated and verified by performing experiments during the LPBF of pure Zn:

(1) The interaction between the scanning laser and metals was numerically modeled, taking the evaporation into consideration of the conservation of energy, momentum, and mass. The influence of the laser energy input and material properties on the evaporation rate was numerically evaluated. The increase in the laser energy input intensified the evaporation, and a higher evaporation led to deeper penetration, a stronger evaporation fume, and more mass loss.

(2) The effect of the shielding gas flow on the evaporation fume was predicted, which provided a powerful tool for the optimal design of a customized gas circulation system. The optimized shielding gas flow efficiently removed the evaporation fume and guaranteed stable melting during the LPBF of pure Zn.

(3) With an adequate laser energy input and an optimal shielding gas flow, pure Zn samples subjected to the LPBF exhibited a high densification. The effect of evaporation on the formation quality of pure Zn was analyzed and compared with that of the TC4 alloy, which helped better understanding of the LPBF processing technologies of Zn and its alloys for the future biodegradable applications.

## Figures and Tables

**Figure 1 materials-14-02677-f001:**
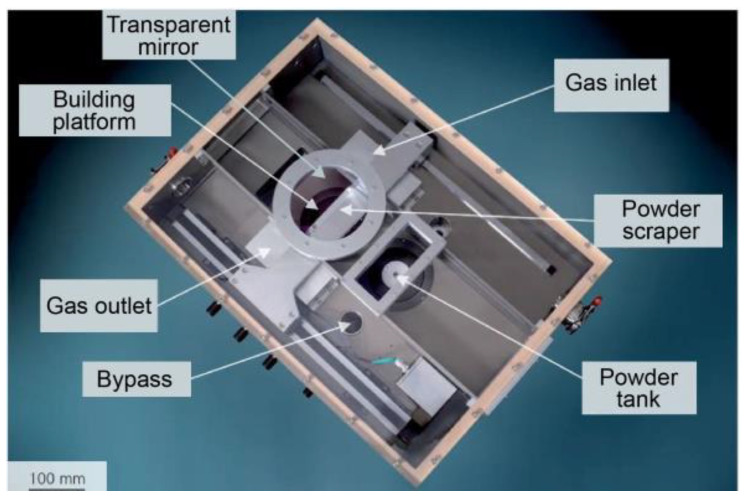
LPBF processing chamber with customized gas circulation system.

**Figure 2 materials-14-02677-f002:**
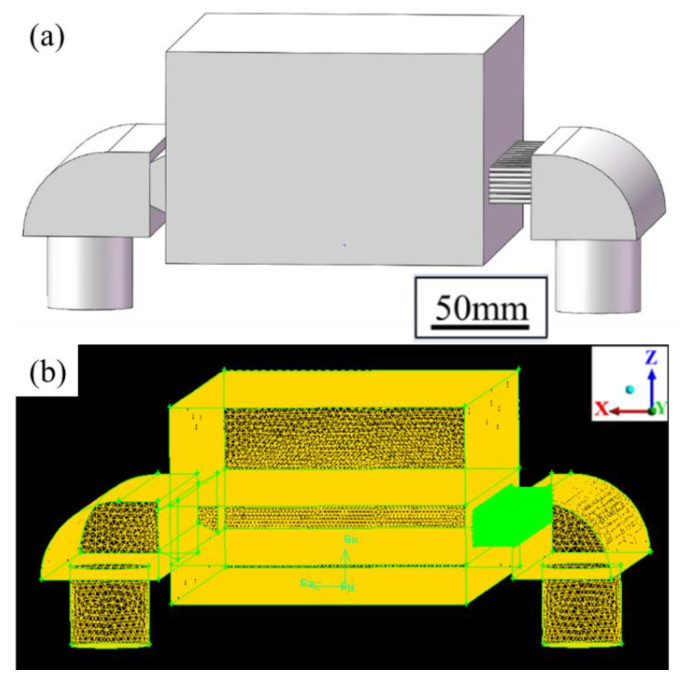
Numerical model of the LPBF processing chamber with a gas circulation system, (**a**) geometrical model; (**b**) after meshing.

**Figure 3 materials-14-02677-f003:**
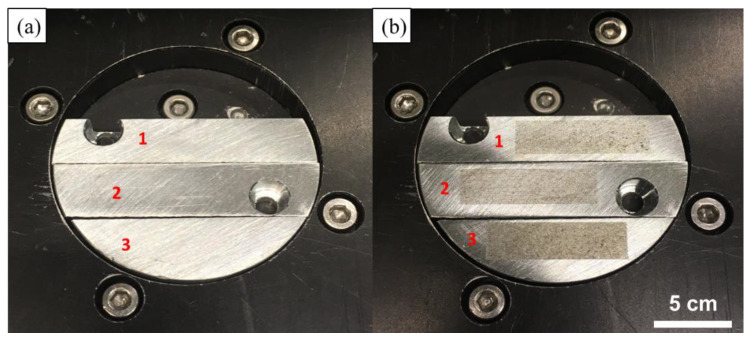
Pure Zn plates (**a**) before laser scanning and (**b**) after laser scanning.

**Figure 4 materials-14-02677-f004:**
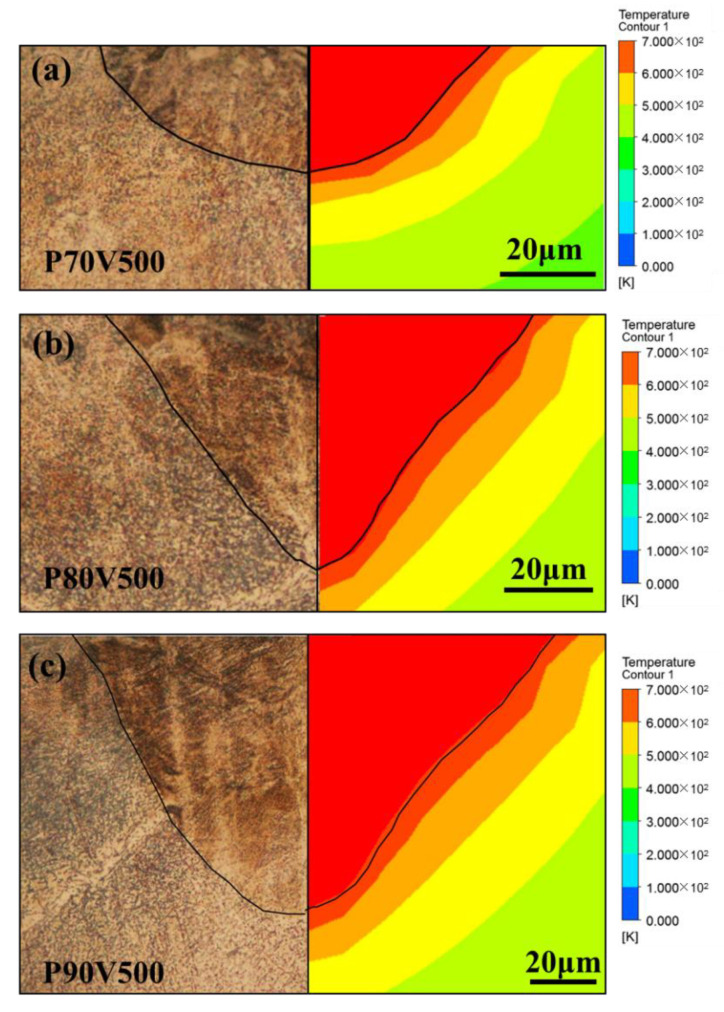
Penetration shape and temperature contour of the Zn plates, as obtained by the experiment and calculation, for laser powers of (**a**) 70 W, (**b**) 80 W, and (**c**) 90 W.

**Figure 5 materials-14-02677-f005:**
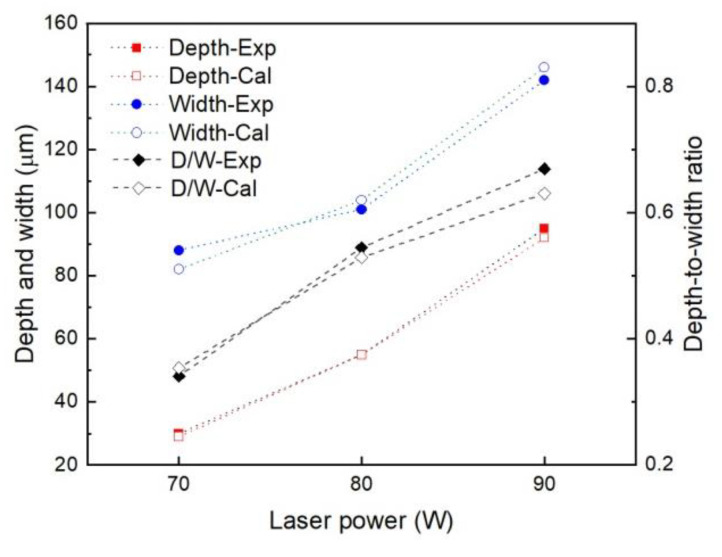
Dimensional values of the penetration shape of pure Zn plates.

**Figure 6 materials-14-02677-f006:**
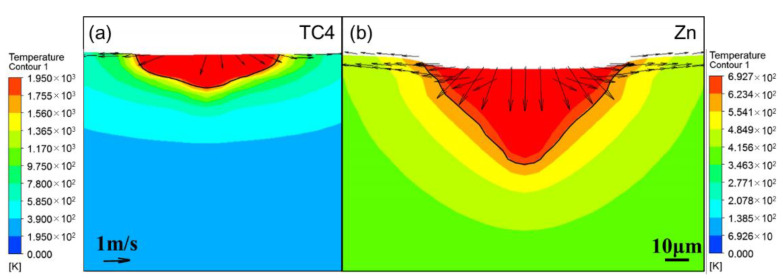
Flow behavior inside the molten pool, as determined by calculation for *P* = 90 W *V* = 500 mm/s, (**a**) TC4, (**b**) pure Zn.

**Figure 7 materials-14-02677-f007:**
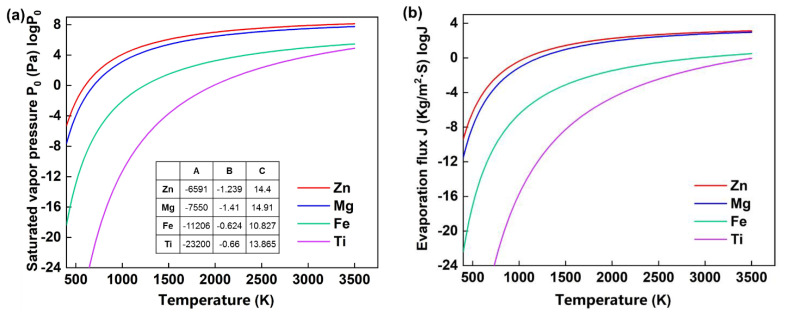
Saturated vapor pressure (**a**) and evaporation flux (**b**) based on the Langmuir model.

**Figure 8 materials-14-02677-f008:**
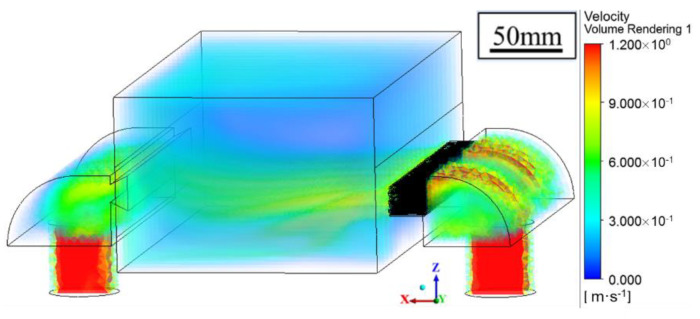
Velocity distribution of shielding gas flow obtained by simulation.

**Figure 9 materials-14-02677-f009:**
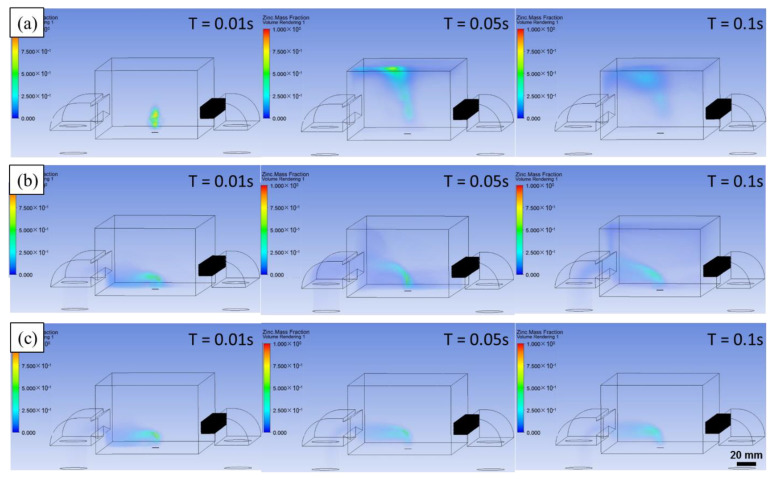
Effect of shielding gas flow on the evaporation fume, (**a**) no shielding gas flow; (**b**) insufficient shielding gas flow; (**c**) adequate shielding gas flow.

**Figure 10 materials-14-02677-f010:**
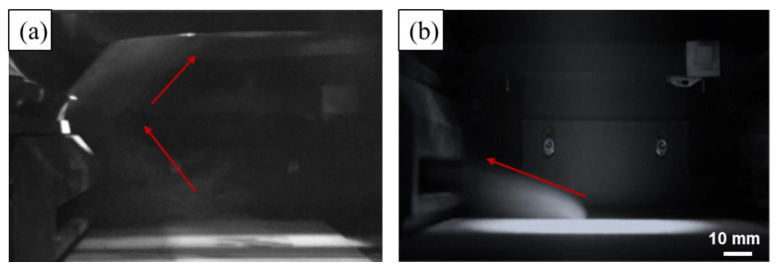
Evaporation fume captured during the LPBF of pure Zn, (**a**) insufficient shielding gas flow; (**b**) adequate shielding gas flow.

**Figure 11 materials-14-02677-f011:**
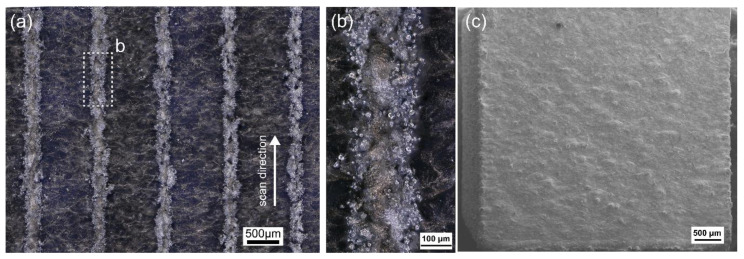
Surface quality of pure Zn by the LPBF; (**a**,**b**) single track, (**c**) top surface of a cube [28].

**Figure 12 materials-14-02677-f012:**
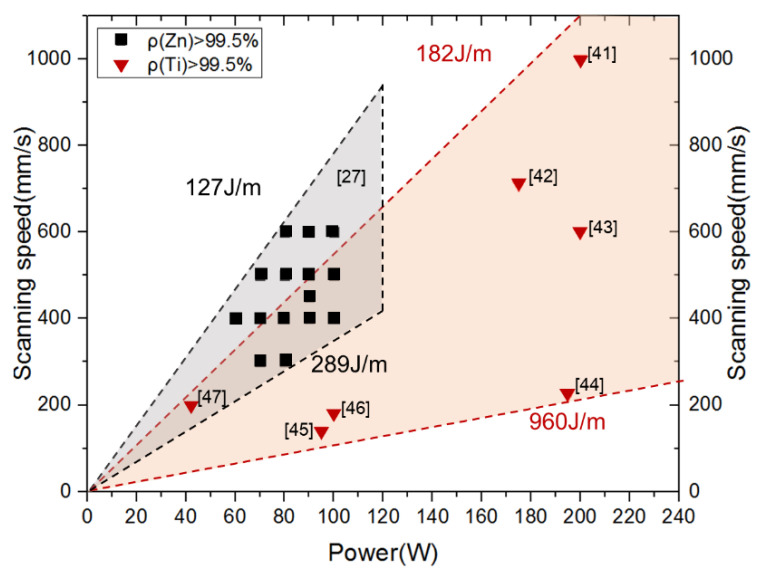
Processing windows of pure Zn and TC4 produced by the LPBF.

**Table 1 materials-14-02677-t001:** Evaporation rate by experiment and calculation.

Material	Laser Power (W)	Mass Loss after Scanning (mg)	Experimental Evaporation Rate (mg/s)	Calculated Evaporation Rate (mg/s)	Deviation (%)
	70	0.4	0.23 ± 0.05	0.225	2.17
Zn	80	0.5	0.29 ± 0.07	0.317	9.31
	90	1.0	0.57 ± 0.10	0.513	10.00

## Data Availability

The data presented in this study are available on request from the corresponding author. The data are not publicly available as the data also forms part of an ongoing study.

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
