# Peer review of "Influence of Laser Energy Input and Shielding Gas Flow on Evaporation Fume during Laser Powder Bed Fusion of Zn Metal"

_materials, 2021, doi:10.3390/ma14102677_

Round 1

Reviewer 1 Report

This manuscript reports numerical modeling of laser powder bed fusion of Zn to predict the laser power-dependent evaporation rate and flow conditions needed to remove evaporated metal. The results are certainly important for optimizing LPBF conditions and therefore are of likely interest to the LPBF community. I have only minor suggestions for improving the clarity of the manuscript as listed below:

  • In the Introduction discussion of "Zn-based metal porous scaffolds", which other metals are used besides Zn? Are these made of alloys?
  • It would be helpful to point out somewhere in the Introduction why Zn has such a high evaporation rate compared to other metals (I assume it's the low melting/boiling point, but it would be good to state this explicitly)
  • The terms "relative density" and "keyhole" on p. 2 in the Introduction are not defined, which may be confusing for a non-expert since these are mentioned throughout the rest of the text
  • In section 2.1, some justification should be given for assumptions (4) and (5) in the laser-metal interaction model
  • On p. 4 right before section 2.2., the text refers to Eq. (12), which doesn't appear until several pages later for a completely different model. Is Eq. (11) meant?
  • What type of laser was used for the experiments in Section 3?
  • What is the physical reason for the significant enhancement of evaporation going from 80 to 90 W? Does the fluence cross a threshold to induce some nonlinear effects?
  • It would be nice to relate the results from the laser-metal interaction model with different laser powers to the gas flow results: would "sufficient" shielding flow need to be higher for higher laser powers?
  • In the Discussion section on p. 11, why is assuming the peak temperature in the molten pool is equal to the boiling point valid? In the previous sentence several works are cited showing a measured temperature higher than the boiling point.

Reviewer 2 Report

The article deals with a very important scientific fact and phenomena. Vapours and fumes generated during the laser powder bed fusion have a negative effect on the consistency of the energy density applied and then the AM part mechanical/physical and chemical properties. the man parameters affecting the generation of the fumes are; the laser beam power and spot size, scanning speed and strategy, inert gas supply metal powder type, powder layer thickness and the build chamber size. These fumes are composed of metal vapour, burned gases and carrying out powder particles and molten material. The negative effect of these can be represented by the interruption of the laser beam path to the metal powder surface and the multi-reflections which might occur. The most significant way to reduce these effects is the inert gas circulation which helps in blowing these fumes away from the melt-pool surface toward the central filtration unit. The gas supply rate is restricted to the deposited powder removal meaning that the latter is a very important factor to be experimentally tested in order to validate any modelling. 

In your study, you adopted a solid Zn plate of 25 mm thickness assuming it is a sufficient replacement for the metal powder and that the results you obtained are accurate. This is incorrect for the reasons mentioned in the report attached. For the above,  I have made my decision to reject your paper.

Regards

Reviewer 3 Report

This is an interesting work, with numerical analyses validated with experimental results. The scope and discussion are interesting to any reader interested in this field.

Nevertheless, I have one remark regarding the element size. The authors do not indicate if a mesh convergence analysis was carried out. This is important to guarantee the results obtained do not depend on the elements size. Nevertheless, the element size indicate is 0.01x0.01x0.01 m3, which results in a FE mesh that can considered refined enough. Therefore, I consider this one a minor issue.

Reviewer 4 Report

For Authors

In this work, the effect of evaporation on the Laser powder bed fusion (LPBF) process was numerically simulated and verified by performing experiments with pure Zn.

The work is interesting for future applications of biodegradable Zn based metals (i.e. metal bone implants).

Thus, I think that this paper may be published in Materials just after the authors explain some issues in more detail. I also add some comments that I hope can help the authors

The authors should explain more precisely:

Comments:

1.Introduction:

Page 3, line 1-2: “…and compared with the LPBF of TC4 titanium alloy”.

Why??. My recommendation is that you change your sentences in page 6 line 29 “ TC4 titanium alloy has been widely used as the material of non-degradable metal implans” to here

2.Numerical simulations and methods:

Page 3, line 16: For clarity, I think it is good tou say what is energy source, force source, mass source. I mean, is by unit of time…

Page 3, line 18: “A solid metal plate, rather than powder bed, was used.. Reasons: Firstly…Secondly. And the presence of a layer of powders results in only a second order effect”

Can you explain it in more detail?. The fact is that experimentally we have powder bed. Please, could you write the validity of this approximation, I mean using solid metal plate instead of power bed will not have an effect higher than 20%...

Page 3, line 26-30: It would be better to use a,b,c,d,e instead of 1,2,3,4,5 for classification (you already are using 1,2,3,4,5 for equations).

Page 3, line 30: For clarity, It would be better that you introduce Knudsen layer in the Introduction section

Page 3, line 31: “ The interaction between metal vapor and the incoming laser beam was omitted”.

Is this aceptable?? .Please, explain it (you said in Introduction, page 2, line 23 “that metal vaporcools down and forms fume”)

Page 4: The equations are out of the margins of the page

Page 4, last lines: “minimum element size was set 1mm….4 mm, in order to get balanced performance between calculation accuracy and efficiency”.

But, you did not write any comparison of your results with 1mm or 4 mmm. Please write some numerical comparison in results

2.Materials and experiments:

Page 6, line 9: Figure 3. Is missing. I do not have it¡¡¡

Page 6: Laser energy is a key parameter in the experiment, but you did not describe anything about the laser and the optics system that you used during the experiment. In the introduction you wrote that the focused laser beam had intensity over 106 W/cm2,and in 3. Materials and experiments you said that laser power was varied (70,80 and 90 W). How do you focus it, what is the wavelength, is it a pulsed laser or a continuous wave laser??.

Page 6, line 29: “ TC4 titanium alloy has been widely used as the material of non-degradable metal implans”.

From my point of view, this is not Results. (better in Introduction page 3 line 1-2

Page 7, at the end: The Knudsen…..better: The Knudsen layer

Page 8: Fig 5: I do not see the D/W-Exp point, is it missing??

Page 9, Table I. Deviation. How do you obtain it?. Did you take into account the statistics from 3 measurements

Page 11, line 46-47.”More work is expected in the future for the numerical simulation of the compositional change during LPBF process”.

However, if it has not been studied, not only numerical simulation would be expected, experimental measurement should be essential…

General comments:

Page 3. You assume that the pressure is 1.07 bar (page 3, line 29). Why?. In the experimental results you did not give the value of the pressure, Why?

From y point of view more discussion between experimental results and numerical calculations are needed

English and writing:___________________________________________________

Page 2, line 10: change acctually to actually

Page 2, line 12: keyholy??? (better keyhole???)

Page 2, line 16: dozens of times (higher than???)

Page 2, line 20: change evaporaion to evaporation

Page 2, line 21: change compostion to composition

Page 2, line 23: change evporation to evaporation

Page 2, line 25: change aborbed to absorbed

Page 2, line 41: change infared to infrared

Page 2, line 43: change evaportion to evaporation

References: Reference 47 is not in the text???
